# Detection of Multiple Mental Disorders from Social Media with Two-Stream Psychiatric Experts

**Siyuan Chen**[1]*, **Zhiling Zhang**[2]*, **Mengyue Wu**[3]†, **Kenny Q. Zhu**[4]†

[1,3] X-LANCE Lab, Department of Computer Science and Engineering
MoE Key Lab of Artificial Intelligence, AI Institute
[1,2,3]Shanghai Jiao Tong University, Shanghai, China
[4]University of Texas at Arlington, Arlington, Texas, USA
{[1]chensiyuan925, [2]blmoistawinde, [3]mengyuewu}@sjtu.edu.cn,
[4]kenny.zhu@uta.edu

## Abstract

Existing Mental Disease Detection (MDD) research largely studies the detection of a single disorder, overlooking the fact that mental diseases might occur in tandem. Many approaches are not backed by domain knowledge (e.g., psychiatric symptoms) and thus fail to produce interpretable results. To tackle these issues, we propose an MDD framework that is capable of learning the shared clues of all diseases, while also capturing the specificity of each single disease. The two-stream architecture which simultaneously processes text and symptom features can combine the strength of both modalities and offer knowledge-based explainability. Experiments on the detection of 7 diseases show that our model can boost detection performance by more than 10%, especially in relatively rare classes. [1]

## 1 Introduction

Mental Disease[2] Detection (MDD) is of great practical value and social benefits, since mental disorders can greatly affect sufferers' life quality (Dreisbach et al., 2019). Lots of practices (Coppersmith et al., 2015; Mowery et al., 2017) indicate that social media posts, containing sufficient expressions about one's feelings and symptoms, can be an informative data source for text-based automatic MDD, which aims to predict whether a person suffers from certain mental diseases.

However, traditional MDD methods (Yates et al., 2017; Trotzek et al., 2020) process every post in the user's posting history, which can include many irrelevant or distracting posts. To avoid these noises, some prior works try to extract key posts by clustering (Zogan et al., 2021) or semantic similarity (Zhang et al., 2022a), but these heuristics can still introduce erroneous posts, affecting the subsequent MDD results.

Moreover, comorbidity of several mental disorders is common (Roca et al., 2009). For instance, 75% of depression patients in the surveyed population also suffer from anxiety disorder in their lifetime (Lamers et al., 2011). Some research (Adam, 2013) further suggests that mental diseases lie along a spectrum, hence it is quite usual for one person to develop symptoms of several related mental disorders at the same time, and be diagnosed with multiple diseases. However, detecting multiple mental diseases simultaneously in the scenario of comorbidity is still under-explored. Most existing works focus on the detection of a single common disorder, such as depression (Losada et al., 2017; Lee et al., 2021), ignoring the frequent comorbidity diagnosed in clinical practice.

To address these limitations, we aim to explore an approach to detect multiple mental disorders simultaneously in a comorbidity dataset, in which the diagnosed users can have one or more disorders. For simplicity, we refer to this detection task as **Multiple MDD** in this paper, and it can be viewed as a multi-label classification problem.

Intuitively, detecting multiple mental diseases can be challenging to resolve, as there are lots of overlapping clinical manifestations shared among different diseases, so the labels implicitly intersect with each other. Pioneering works on multiple MDD (Cohan et al., 2018; Sekulic and Strube, 2019) commonly yield unsatisfying results as they simply use a shared model architecture for all diseases, which may not be strong enough to distinguish between diseases. Zhang et al. (2022b) shows the effectiveness of implementing psychi-

---

*These authors made equal contribution.

†Corresponding authors. This work has been supported by National Natural Science Foundation of China (No.61901265), Shanghai Municipal Science and Technology Major Project (2021SHZDZX0102), Key Research and Development Program of Jiangsu Province (No.BE2022059-2) and Alibaba Innovative Research.

[1]Code is available at https://github.com/chesiy/EMNLP23-PsyEx. Dataset can be provided upon request.

[2]In this work, we will use 'mental disorder' and 'mental disease' interchangeably.

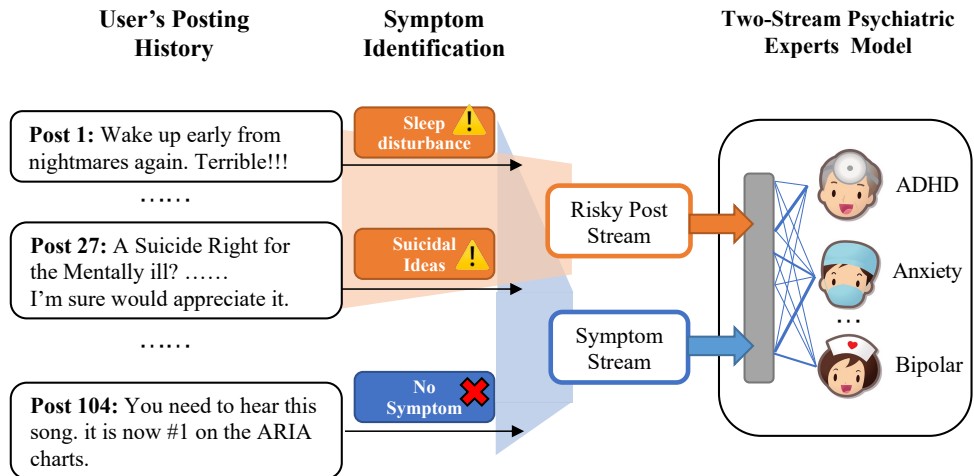

**User's Posting History** · **Symptom Identification** · **Two-Stream Psychiatric Experts Model**

Figure 1: Psychiatric Experts Model with Symptom-based Risky Post Screening. Only posts highly related to psychiatric symptoms will be selected as diagnostic basis for further mental disease detection, which forms the risky post stream (the orange part). The symptom stream of a user contains all the symptom identification results from his/her whole posting history (the blue part), providing a more global view for the MDD model.

atric symptom knowledge on multiple MDD, which is more interpretable and can outperform previous pure-text methods due to its better clinical grounding. However it detects each disease separately, ignoring the inner correlations between diseases, leading to unsatisfying results on rarer diseases like OCD.

In this work, we propose **PsyEx** (**Psy**chiatric **Ex**perts), a multi-task learning framework that can simultaneously detect 7 mental disorders[3] with a shared backbone and disease-specific structures to leverage the common characteristics of all diseases while still being able to capture their distinctions. This framework consists of two phases (see Figure 1). First, we utilize a symptom identification model that explicitly leverages psychiatric knowledge to obtain symptom features for the *symptom stream*. The symptom features are also used to select posts that show high symptom risks. Then another *risky post stream* implicitly learns clues for mental diseases from the semantics of the selected posts[4]. Therefore, these two streams can complement each other and improve interpretability based on domain knowledge. They are fed into the multiple MDD model which has 7 disease-specific psychiatric experts on top of a shared hierarchy network, so that the backbone can learn the shared knowledge be-

tween diseases, while each expert can focus on the characteristics of each specific disease.

Experiments show that our method can achieve SOTA multiple MDD performance across 7 diseases and bring significant improvement to rarer diseases on which the baselines struggled heavily. Our main contributions are:

- We successfully exploit psychiatric symptoms in this multiple MDD framework, including the symptom-based screening to facilitate a precise selection of risky posts, and the symptom stream to provide domain knowledge, together with a more holistic view of the entire posting history.

- We propose a two-stream Psychiatric Experts Model for better multi-task learning of 7 mental disorders, which boosts the overall performance by better utilizing the distinctions and commonality among diseases.

- With the interpretability enabled by symptom and disease-specific experts, we analyze the decision-making process of PsyEx, and further study the contribution of each symptom to the detection of different diseases.

## 2 Approach

In this section, we introduce the proposed framework for multiple MDD, including symptom identification and a two-stream Psychiatric Experts Model. First, we give the definition of multiple

---

[3]The 7 mental disorders are: ADHD, Anxiety Disorder, Bipolar Disorder, Depression, Eating Disorder, OCD and PTSD. Brief introductions about these disorders are included in Appendix A.

[4]We call them "streams" here because both symptom features and text features are arranged in chronological order.

MDD here. Given the posting history of a user with N posts: $\{p_1, p_2, ..., p_N\}$, and a list of $M$ potential mental diseases, the goal is to detect which disease(s) in the $M$ candidates does the user suffer from. Typically, this can be accomplished by training $M$ separate models for each disease. Here, we further propose a novel method that leverage a single model to effectively learn the distinctions and commonality among diseases for a better overall performance on all diseases.

## 2.1 Symptom-based Risky Posts Screening

Traditional MDD method processes every single post equally overlooking the fact that not every post from a patient reveals useful information for detection. To facilitate the MDD model performance, as well as provide explainable diagnosis basis, we screen risky posts first, in particular with disease-dependent symptom information, and use the selected posts for multiple disease detection.

Inspired by clinical diagnosis procedures, we assume that posts reflecting *symptoms* of mental disorders suggest higher risks. Since healthy individuals rarely produce symptom-related content, posts with symptom indications could better separate patients from control users. Hence, unlike the prior heuristic methods discussed in Sec. 1, we implement a screening method based on the symptom features extracted by a supervised symptom identification model[5] (Zhang et al., 2022b). This model can identify 38 symptom classes from 7 mental diseases with a Mental BERT-based encoder (Ji et al., 2022) and a linear classifier. It is trained on a large-scale, multi-disease annotated symptom identification dataset, based on the globally-accredited diagnostic criteria from Diagnostic and Statistical Manual of Mental Disorders (DSM-5) (APA et al., 2013).

The symptom feature of each post is a 38-dimensional vector, where each dimension is the predicted probability of a certain symptom [6]. We can then estimate the *risky score* of a post with the *sum* of predicted probability among all symptoms. The top $K$ posts with highest risky score among a user's whole posting history (containing $N$ posts) will be selected as his/her risky posts. In practice,

we set $K \ll N$, so the reduced input size enables high efficiency even with BERT-based language models in the following disease detection phase. Moreover, the symptom extracted by a supervised model is also more reliable to select risky posts than heuristic approaches, and we will show this in the experiments (Sec. 3.4.3).

## 2.2 Two-Stream Psychiatric Experts Model

Although MDD can be formulated as a typical text classification problem, traditional model structures are not appropriate for our multiple mental disease detection task for two primary reasons. On the one hand, employing a single user representation to detect all diseases would overlook the explicit differences between them. Conversely, if each disease is detected separately, the neglect of shared symptoms across these diseases will also adversely affect the overall performance.

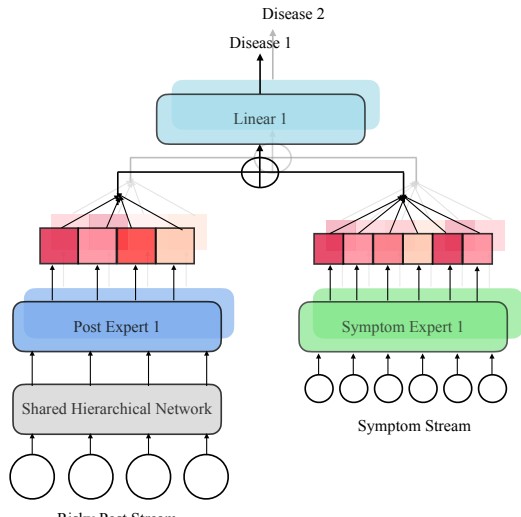

Figure 2: Illustration of the proposed model architecture, with 2 disease-specific experts on top of a shared network. Darkness of red square indicates attention strength, which is different for each disease.

To alleviate this problem, we propose a Two-Stream Psychiatric Experts Model (See Figure 2), which can detect all the mental diseases simultaneously through multi-task learning while still capturing the nuances among them. It takes *risky post stream* and *symptom stream* obtained in symptom identification phase as input to combine the advantages of both modalities.

**Risky Post Stream Model** The risky post stream "cherry-picks" $K$ posts with highest symptom probability, providing more strong and concentrated signals of mental diseases. To better explore these sig-

---

[5]The code and data of the symptom identification model can be found at https://github.com/blmoistawinde/EMNLP22-PsySym

[6]These symptoms (e.g., *anxious mood, sleep disturbance, poor memory*) are carefully extracted from DSM-5, so that there is as little semantic overlap as possible between them. We list all the symptoms in Table 12 (Appendix E).

nals and distinguish among multiple diseases, the model consists of two components: a single **shared hierarchical network** and $D$ **disease-specific post experts**.

In the shared part, we draw on the structure of Hierarchical Attention Network (HAN) (Yang et al., 2016) with post and user-level encoder to make better use of the sequential structure of the posts. We employ a pre-trained BERT model as the post encoder. For words $\{w_1, w_2, ..., w_L\}$ in a post, the post representation $p$ is,

$$p = BERT_{[CLS]}(w_1, w_2, ..., w_L) \qquad (1)$$

The user-level encoder utilizes a transformer (Vaswani et al., 2017) structure, modeling the relations between these posts $\{p_1, p_2, ..., p_K\}$, and produces updated post representations $\{p'_1, p'_2, ..., p'_K\}$.

In the disease-specific part, each disease $d$ has its own attention layer to get different attention distributions on the same post sequence. As such, these $D$ attention layers can be considered as feature selectors from the shared network (Liu et al., 2019). Then we perform a weighted sum of the post representations according to attention score to get the distinctive user representations $u_d$.

$$\alpha_{k,d} = \frac{exp(W_d p'_k + b_d)}{\sum_{k'=1}^{K} exp(W_d p'_k + b_d)} \qquad (2)$$

$$u_d = \sum_{k=1}^{K} \alpha_{k,d} p'_k \qquad (3)$$

where $W_d$ and $b_d$ are both learnable parameters specifically for disease $d$.

The attention distribution on the post sequence reflects the weight of each post in the final prediction. Therefore, the attention score can offer interpretability of the model decision, which will be further discussed in Sec. 3.6

**Symptom Stream Model** Symptom feature is critical for MDD, because it equips the detection model with explicit psychiatric knowledge. The symptom stream can be considered as an $N \times 38$ matrix, revealing a user's whole posting history in a lighter way, which is able to replenish the incomplete text stream and provide holistic information for the detection model. To better capture the unique features of different diseases, the symptom stream model is totally disease-specific, and its structure is the same as the corresponding part

in risky post stream model. Similarly, each disease has its own "expert" to get different attention distributions on the same symptom sequence, which is intuitive because each mental disease has its own typical or unique symptoms for diagnosis (e.g., panic fear for Anxiety, intrusion for PTSD).

Finally, the disease-specific user representation from both stream are concatenated and fed into separate linear layers to get the binary predictions on whether the user suffers from a certain mental disease.

The whole model is trained with the standard binary cross entropy loss, where the loss of all the tasks are averaged. In the dataset, we usually have patients that are certain for the diagnosis of some diseases but uncertain for others, and we should not assume these diseases to be not existing due to the high comorbidity of diseases. Therefore, we implement *loss masking* as described in Fonseca et al. (2020), so that for patients with at least one disease, we do not consider their absent disease labels as negative (i.e., we assign zero weight to exclude them from affecting the training loss). This approach helps alleviate the problem of potentially missing labels. We can get more reliable negative labels from control users who show no signals of mental issues.

## 3 Experiments

In this section, we conduct experiments to (1) examine the performance of our two-stream PsyEx model compared with strong baselines; (2) validate the effectiveness of various design choices with ablation tests; (3) evaluate the model's generalizability on different datasets; (4) analyze the interpretability enabled by symptoms and disease-specific "experts".

### 3.1 Multiple MDD Datasets

We construct a multiple MDD dataset by reimplementing the data collection method of SMHD (Cohan et al., 2018). Users and posts were extracted from a publicly available **Reddit** corpus. We select diagnosed users by detection patterns with a focus on high precision. The patterns consist of two components: one that matches a self-reported diagnosis (e.g., "diagnosed with"), and another that maps relevant keywords to the 7 mental diseases (e.g., "panic disorder" to Anxiety). A user is assigned a disease if one of its keywords occurs within 40 characters of the diagnosis pattern. Control users (i.e.,

| Method | Depression | Anxiety | ADHD | Bipolar | OCD | PTSD | Eating | Avg. F1 |
|---|---|---|---|---|---|---|---|---|
| TF-IDF+LR (Cohan et al., 2018) | 76.75 | 75.45 | 68.97 | 76.56 | 40.0 | 44.02 | 30.0 | 58.82 |
| BERT (Nguyen et al., 2022) | 73.28 | 70.59 | 56.72 | 61.21 | 41.82 | 50.0 | 32.67 | 55.18 |
| Symp (Zhang et al., 2022b) | 81.06 | 81.99 | 70.3 | 81.75 | 65.12 | 64.41 | 61.54 | 72.31 |
| HAN-GRU (Sekulic and Strube, 2019) | 74.99 | 82.16 | 81.72 | 80.28 | 70.59 | 67.67 | 68.57 | 75.14 |
| ChatGPT | 70.12 | 73.09 | 64.08 | 67.12 | 52.98 | 67.61 | 29.73 | 60.68 |
| PsyEx | **87.89** | **89.84** | **84.40** | **91.58** | **81.69** | **81.69** | **85.71** | **86.12** |

Table 1: Mental Disease Detection Results across 7 diseases on Reddit dataset, reporting F1 scores in binary setting. We order these diseases in descending order according to their number of patients in the dataset, making it easier to identify rarer classes.

| Method | Depression | Anxiety | ADHD | Bipolar | OCD | PTSD | Eating | macro-F1 | micro-F1 | EM |
|---|---|---|---|---|---|---|---|---|---|---|
| TF-IDF+LR | 52.16 | 32.52 | 29.9 | 43.14 | 13.79 | 9.76 | 0 | 25.9 | 38.09 | 77.37 |
| BERT | 56.99 | 47.03 | 31.86 | 16.59 | 0 | 0 | 0 | 21.78 | 39.38 | 75.32 |
| Symp | 62.99 | **57.93** | 49.57 | 51.85 | 18.18 | 0 | 0 | 34.36 | 53.05 | 76.65 |
| HAN-GRU | 59.4 | 44.88 | 53.81 | 62.2 | 0 | 0 | 0 | 31.47 | 50.47 | 77.55 |
| ChatGPT | 51.48 | 46.3 | 41.83 | 62.45 | 28.72 | 20.54 | **12.12** | 37.64 | 44.16 | 75.56 |
| PsyEx | **66.09** | 54.97 | **54.07** | **66.17** | **41.51** | **44.83** | 9.52 | **48.17** | **58.59** | **81.38** |

Table 2: Mental Disease Detection Results across 7 diseases on Reddit dataset in mutli-label setting, reporting F1 score of each disease, micro-F1 and macro-F1 over all diseases, as well as exact match ratio (EM).

healthy persons) are randomly sampled from those who never posted or commented in mental health related subreddits and never mentioned the name of 7 mental diseases (e.g., "bipolar", "PTSD") to avoid possible false positives. Similar to SMHD, we eliminate the diagnostic posts from the dataset to prevent the direct leakage of label, but retain those mental health related posts to allow the extraction of symptom-related features.

The final dataset consists of 5,624 diagnosed users and 17,209 control users. Each diagnosed user can have one or more disease labels, so we provide the distribution of the user's disease count in Table 8 (Appendix B). The statistics show that 57% users in the dataset suffer from two or more kinds of mental disorders, and this comorbidity scenario is challenging for disease detection models. Moreover, due to the uneven distribution of different mental disorders in reality, the dataset is naturally unbalanced, with more users suffering from Depression and Anxiety, while fewer users with OCD and PTSD[7]. Consequently, the detection of rarer diseases can be even more difficult to resolve.

### 3.2 Methods of Comparison

For multiple MDD task, we mainly compared the proposed methods with 4 types of baselines: **TF-IDF+LR** (Cohan et al., 2018) is a representative traditional machine learning method which utilizes TF-IDF to extract textual features, followed by a Logistic Regression model for prediction. **BERT** is the reimplementation of the MDD model in Nguyen et al. (2022), which utilizes CNN of various kernel sizes on top of the sentence embeddings from pre-trained BERT as features to aggregate the information from user posting list. **Symp** (Zhang et al., 2022b) uses the same CNN backbone, but further replace the BERT embedding features with symptoms features, and it establishes a strong baseline. **HAN-GRU** reimplements the hierarchical attention network for MDD proposed in Sekulic and Strube (2019), which utilizes Bidirectional GRU as encoders. Since large language model shows superior performance in various NLP tasks (Ye et al., 2023), we also incorporate such systems for comparison, and uses **ChatGPT**[8] to predict user diseases. More details like hyperparameter settings, prompts of the baseline experiments can be found in Appendix C.

### 3.3 Experimental settings

For PsyEx model, we select $K = 16$ high-risk posts during the screening process to form the risky post stream[9]. We utilize pre-trained bert-tiny[10] (in binary setting) or mental-bert[11] (in multi-label setting) as the basis of the post encoder. The user encoder is a 4-layer 8-head transformer encoder. We train with a batch size of 32 and set learning

---

[7]See Table 7 in Appendix B for the exact number of users suffering from each disease.

[8]https://chat.openai.com/

[9]We further explore the impact of post number on the detection results in Figure 5 (Appendix D).

[10]https://huggingface.co/prajjwal1/bert-tiny

[11]https://huggingface.co/mental/mental-bert-base-uncased

rate at $1e^{-5}$. We also employ early-stopping with a patience of 4 epochs according to validation performance to prevent overfitting.

## 3.4 Experiment Results

To be consistent with previous works (Cohan et al., 2018), we train and evaluate the models in both binary and multi-label setting.

### 3.4.1 Binary Setting Results

For the binary task, we only need to decide whether the user is suffering from a certain mental disease, so only users with this mental disorder plus all control users are selected to train and evaluate. The results in binary setting are shown in Table 1. We can see that our proposed *PsyEx* outperforms all the baseline methods including *ChatGPT*, suggesting the advantage of our symptom-based risky post screening and two-stream psychiatric expert model. Further, owing to the multi-task learning and the shared knowledge between diseases, the detection effect of rarer classes (i.e., eating disorder, OCD, PTSD) is largely improved.

### 3.4.2 Multi-label Setting Results

In multi-label setting, we have to determine if and which mental diseases the user was diagnosed with, that is, the user can have one or more diseases, and all data is used for both training and evaluation. We show the multi-label results in Table 2, in which we evaluate these classifiers with a strict metric, exact match ratio, together with macro and micro F1 score which take partially correct into consideration.

This setting is challenging and underexplored in previous works mainly due to the complexity brought by comorbidity, as well as the various overlapped manifestations among different mental disorders. Our PsyEx model shows significant advantage over other strong baselines, especially on the rarer classes, in which some classifiers can not even find a true positive sample. It is worth noting that ChatGPT also exhibits acceptable performance on rare classes, benefiting from its training on an extensive amount of data, which enhances its robustness and ability to generalize to infrequent cases. However, its practical usage in the mental health domain is limited by its slow speed and high resource requirements.

### 3.4.3 Ablation Study

We examined the effectiveness of various design choices of the proposed Two-Stream PsyEx model

with ablation tests in binary setting. Results are shown in Table 3.

| Method | Avg. F1 |
|---|---|
| Two-Stream PsyEx | **86.12** |
| *w/o* symp-stream | 84.67 |
| *w/o* multi-attn | 83.68 |
| *w/o* multi-task | 85.40 |

Table 3: Ablation tests for the design choices of PsyEx, reporting average F1 across 7 diseases on Reddit dataset. Results of each disease are in Table 9 (Appendix D).

First, we implement a **w/o symp-stream** model[12], which only preserves the risky post stream part to exhibit the effectiveness of symptoms. As shown in Table 3, the detection performance drops without symptom stream, indicating that symptoms can not only disentangle multiple diseases better with embedded domain knowledge, but also provide a global view of users' entire posting history.

Then, we examine the $D$ disease-specific attention layers by implementing a model **w/o multi-attn**, in which all the diseases share a single attention head but are still trained simultaneously with multi-task learning and both streams are preserved as well. We can see that the performance is greatly harmed without multiple attention heads, illustrating the effectiveness of disease-specific "experts" to properly capture the characteristics of different mental diseases.

Moreover, the **w/o multi-task** method further trains a *w/o multi-attn* model for each disease separately without multi-task learning. We can notice a slight decrease on the detection performance even with much more model parameters, as we need to train 7 independent models for *w/o multi-task* method.

| Screening Method | Avg. F1 |
|---|---|
| Symptom-based (PsyEx) | **86.12** |
| Similarity (Zhang et al., 2022a) | 82.47 |
| K-Means (Zogan et al., 2021) | 74.76 |
| Last | 56.74 |

Table 4: Ablation test for different risky post screening methods, reporting F1 score averaged across 7 diseases. Results of each disease are in Table 10 (Appendix D).

We also compare our symptom-based risky post screening method with other approaches in the literature. **Similarity** and **K-Means** both utilize a pre-trained Sentence BERT (Reimers and Gurevych,

---

[12]"w/o" means without, so symptom stream is removed in this model.

| Method | Depression | Anxiety | ADHD | Bipolar | OCD | PTSD | Avg. F1 |
|---|---|---|---|---|---|---|---|
| tf-idf+LR | 36.5 | 37.33 | 61.19 | 56.01 | 3.8 | 8.5 | 33.89 |
| BERT | 56.3 | 53.43 | 57.86 | 62.38 | 44.24 | 45.57 | 53.30 |
| Symp | 41.49 | 37.93 | 31.24 | 41.52 | 32.49 | 22.46 | 34.52 |
| HAN-GRU | 72.24 | 71.21 | 65.17 | 75.44 | 58.58 | 60.24 | 67.15 |
| **PsyEx** | **76.51** | **77.31** | **67.15** | **78.36** | **67.35** | **65.64** | **72.05** |

Table 5: Mental Disease Detection Results across 6 diseases on Twitter dataset, reporting F1 scores in binary setting.

2019) to obtain sentence representations. The former one extracts key posts according to the cosine similarity between post and mental disease descriptions, and select $K$ posts with the highest similarity score. The later one runs K-means clustering algorithm on the post embeddings and gets the $K$ posts nearest to the cluster center as representative posts. **Last** simply selects the *last $K$* post as risky posts. Except for the different screening methods, we use the same model structure (i.e., our proposed PsyEx), and experiment results are listed in Table 4. It can be seen that symptom-based risky post screening outperforms all the heuristic approaches, especially *K-Means* and *Last*, highlighting the importance of a precise screening method, as there can be large amount of posts irrelevant to mental disorders in the users' posting history.

### 3.5 Generalizability

To evaluate the models' generalizability, we conduct experiments using another dataset (Suhavi et al., 2022a) sourced from Twitter. Despite differences in language style and topics, Twitter posts generally feature more shorter content and higher frequency compared to Reddit posts.

This Twitter dataset includes users with eight distinct mental disorders, some of whom have comorbidity. Notably, it has a 60% hand-annotation rate, ensuring the precision of disorder labels. We worked with a subset of this dataset, and for detailed data statistics, please refer to Appendix B.

We conduct experiments on this dataset in binary setting. Given the shorter yet more frequent nature of tweets, we select $K = 128$ tweets as risky posts, and the other hyper-parameter settings (Refer to Sec. 3.3 and Appendix C for details) remain consistent with the Reddit dataset without further tuning. The experimental outcomes are in Table 5. These results demonstrate the effectiveness of our proposed PsyEx model on Twitter data, surpassing the performance of all baselines, which illustrates the model's capability to generalize across various social media platforms.

### 3.6 User-level Interpretability Analysis

The interpretability of our proposed PsyEx model primarily stems from two key factors: (1) the usage of symptoms in risky posts screening and as direct input for model prediction; (2) the incorporation of disease-specific "experts" through the attention mechanism, which allows us to assess the importance of each post and symptom in determining the final prediction, providing insights into which posts and symptoms have the most significant influence on the model's decision.

Consequently, the interpretability of PsyEx can be manifested in two aspects: **Difference**, different symptoms can contribute differently to predicting different diseases; **Reasonableness**, the contribution proportion of symptoms to different diseases in the model is reasonable (i.e., is roughly consistent with the authoritative DSM-5 criteria).

Therefore, we provide a concrete user-level example to illustrate the decision-making process of PsyEx. The selected user suffers from three mental disorders, including *anxiety, bipolar* and *depression*. We apply the trained PsyEx model to his/her posting history, and obtain the attention score matrix of symptom stream, which is $D \times N$, since there're $D$ disease-specific attention heads. To figure out which symptoms are more critical for diagnosing a certain disease, we measure the *contribution* of each symptom to the detection of different diseases, with the help of attention score matrix.

For each diagnosed disease $d$, we calculate the **symptom contribution vector** $C_d$ as follows. First, we select eight symptom probability vectors $S_d = \{s_1, s_2, ..., s_8\}$ with the highest attention score among the symptom stream. Next, for each 38-dimensional vector $s_i$, we only preserve the value of three symptoms with highest probability. So we set the probability of rest symptoms to 0 and obtain $\hat{s}_i$. Finally, we can get the user's symptom contribution vector

$$C_d = \sum_{i=1}^{8} \hat{s}_i \qquad (4)$$

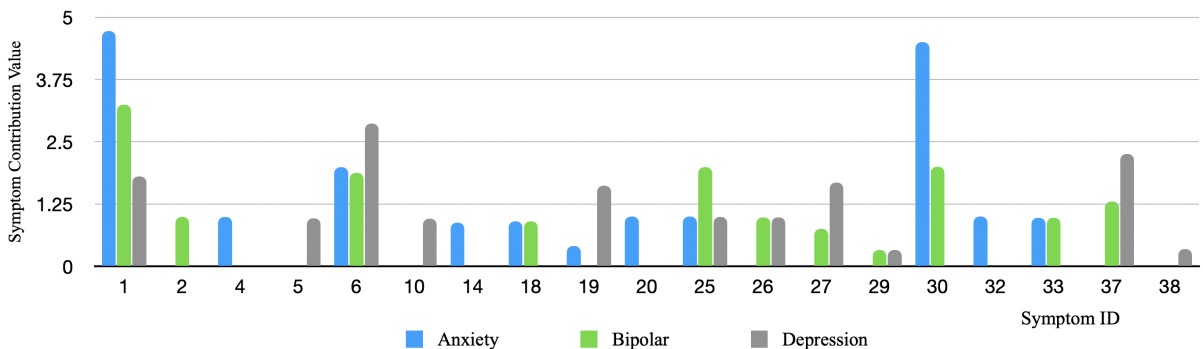

Figure 3: User-level symptom contribution values according to the attention scores obtain from symptom stream model. The user is diagnosed with anxiety, bipolar and depression. We provide the corresponding symptom names with symptom ID in Table 12 (Appendix E), and we omit the symptoms whose contribution values of anxiety, bipolar and depression are all 0.

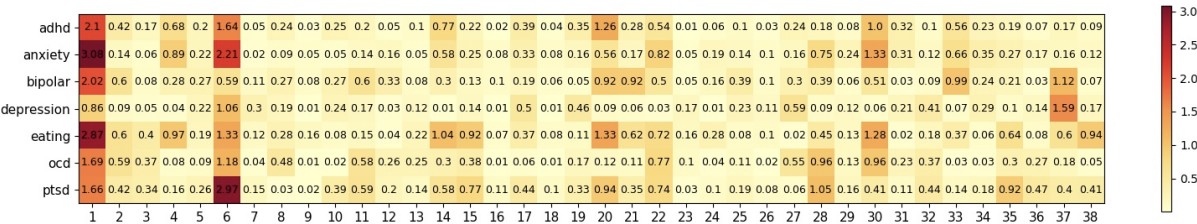

Figure 4: Global symptom contribution vectors of all users in the test set. We provide the corresponding symptom names with symptom id in Table 12, Appendix E.

which is demonstrated in Figure 3.

The symptom contribution vectors of these three mental diseases are quite different from each other, satisfying the first aspect of interpretability (i.e., Difference), which means that our proposed model, to some extent, have learned how to "diagnose" a certain disease with its corresponding symptoms just like psychiatrists.

But do these learned high-contribution symptoms truly make sense for the diagnosis? We compare our symptom contribution vectors with the standard criteria DSM-5 to validate the second aspect of interpretability (i.e., Reasonableness). From Figure 3, we can find out that typical symptoms "6: depressed mood"[13] and "37: weight and appetite change" of depression, "1: anxious mood" and "30: panic fear" of anxiety do contribute more to the detection of them than to other diseases, which is in line with DSM-5 . Moreover, we can also notice many shared high-contribution symptoms between bipolar disorder and depression, which is reasonable because bipolar disorder contains both depressive episodes and hypomanic episodes[14]. There-

fore, we claim that the explicit usage of symptoms, as well as disease-specific attention layers, can exactly improve the interpretability of our neural network model.

### 3.7 Global Symptom Contribution Analysis

To provide a global view of high-contribution symptoms for all the 7 diseases, we plot a heatmap (See Figure 4) of symptom contribution vectors based on all the users in the test set. Here, we group the users according to his/her diagnosed diseases, and obtain 7 disease-specific user subsets [15]. For each disease subset, we calculate all the user-level symptom contribution vectors as Eq. (4) based on the attention score, and aggregate them by averaging to get the global contribution vector of each disease.

Apart from a lot of agreement reached between our global contribution vector and the authoritative DSM-5 (e.g., "21: drastical shift in mood and energy" for bipolar disorder), we can find several interesting inconsistencies. For example, eating disorder patients are "38: more irritable" and tend to "20: do things that easily get painful consequences", both of which are typical symptoms for bipolar disorder patients in the manic episodes, probably

---

[13]We present the symptoms in "ID: name" format.
[14]https://www.nimh.nih.gov/health/topics/bipolar-disorder

[15]a user with multiple mental diseases is in multiple subsets

due to the high comorbidity of these two disorders proved by previous researches (Lunde et al., 2009; Ruiz and Gutiérrez-Rojas, 2015). Further, almost all the diseases treat "1: anxious mood" and "6: depressed mood" as the most important symptoms in PsyEx, while in DSM-5, they're not the typical symptoms for diagnosing many mental diseases such as ADHD. We try to explain this as follows. As a diagnostic criteria, DSM-5 tend to focus on the distinctive symptoms of each disease rather than the common ones. However, these common symptoms can be generally meaningful to distinguish patients of most disorders from control users, so they are highly weighted by our PsyEx model.

## 4   Related Work

In the literature, substantial efforts have been made to detect a certain mental disease. Some of these works focus on leveraging features like TF-IDF, LIWC (Pennebaker et al., 2001), and posting patterns (Trotzek et al., 2020; Losada and Crestani, 2016) for MDD. Others apply various deep learning methods (Yates et al., 2017; Gui et al., 2019), as well as the contextualized embedding (Ji et al., 2022; Jiang et al., 2020) to improve the performance of classifiers. However, these methods often fail to generalize well (Harrigian et al., 2020) and cannot provide explainable results due to lack of knowledge in the psychiatric domain. To tackle these issues, some works (Lee et al., 2021; Nguyen et al., 2022) began to utilize symptom features, but they extract symptom features with unsupervised/weakly supervised methods, which isn't so reliable for the downstream MDD task.

Recent years, some works start to detect multiple mental disorders. Cohan et al. (2018) proposed a massive Reddit dataset *SMHD* containing 9 mental disorders, followed by many subsequent studies based on this dataset, such as Sekulic and Strube (2019) and Zhang et al. (2022b). However, these works often directly apply a single-disease model to the multi-disease data (i.e., train the model for $D$ times to obtain the results of $D$ diseases), overlooking the correlation among multiple diseases, and thus fail to perform well on rarer diseases.

With the emergence of large language models (LLMs), there exists a few studies utilizing LLMs for tasks like depression detection on social media (Lamichhane, 2023; Qin et al., 2023) or developing chatbots for depression diagnosis (Chen et al., 2023), but we fail to find any existing work that simultaneously detects multiple mental disorders or for mental health symptom extraction. While we acknowledge LLM's potential in mental health application, it needs further exploration and investigation as our preliminary results indicated that this remains a challenging task.

## 5   Conclusions

In this work, we tackle the challenge of detecting multiple mental diseases simultaneously in the scenario of comorbidity, and propose a multiple MDD framework achieving SOTA performance in both binary and multi-label settings. We first apply risky post screening based on symptoms, providing reliable diagnostic basis for further disease detection. Then, we propose a two-stream Psychiatric Experts Model with a shared hierarchical encoder and disease-specific attention layers, which simultaneously process the symptom and text features to combine the advantage of both modality. We also explore the interpretability of PsyEx by providing a user-level analysis, and measuring the global symptom contribution to the detection of different diseases.

## 6   Ethical Statement

We rely on publicly available Reddit posts[16] in our work and we make every possible effort to minimize the risk of leaking privacy of individuals in the data collection process. We made no attempt to contact users or link users to other social media accounts following the previous best practices (Cohan et al., 2018; Losada et al., 2017; Suhavi et al., 2022b). We also replace usernames with random identifiers to prevent users' identities from being known without the use of external information. For the usage of symptom identification dataset, we sign and comply with the data usage agreement to prevent the invasion of privacy or other potential misuses. What's more, we carefully consider the application of mental disease detection from social media. The purpose of this work are not to replace psychiatrists. Instead, we expect our model to be used as an effective auxiliary tool by experienced psychiatrists in the future.

---

[16]In the privacy policy of Reddit, the posts are public and accessible to everyone, and Reddit allows third parties to access public Reddit content via the Reddit API and other similar technologies. See https://www.reddit.com/policies/privacy-policy for more detailed information.

# 7 Limitations

Our work has some limitations that could be addressed in future research.

- Despite the significant performance boosting over the baseline, our proposed PsyEx model still cannot achieve satisfying performance in multi-label setting, especially on rarer diseases like eating disorder (See Table 2). To tackle this issue brought by the imbalanced data, we utilize a commonly-used resampling method, which samples equal amount of users with each disease for each batch. However, we find no improvement in the detection performance of these rarer diseases after balanced sampling, indicating that the unsatisfying results aren't just a matter of sparse positive samples. Therefore, we hope to further address this issue in future studies.

- Multiple MDD task is still under-explored currently. Many previous works (e.g., Sekulic and Strube (2019)) only conduct experiments on binary setting (i.e., separately train $D$ models for detecting $D$ mental diseases). Therefore, for comparison under multi-label setting, we can only adopt their hyperparameters on the binary setting, which may not be optimal in some cases.

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

## A Introduction of 7 Mental Disorders

Here, we briefly introduce 7 mental disorders studied in our paper, listing their typical manifestations summarized from DSM-5 in Table 6 for better understanding of these mental disorders.

| Disease | Typical Symptoms |
|---------|------------------|
| ADHD | inattention; hyperactivity and impulsivity. |
| Anxiety | excessive fear and worry; panic attacks; anxious mood. |
| Bipolar Disorder | drastic shift in mood and energy; experience periods of mania and depression. |
| Depression | depressed mood; loss of pleasure or interest; poor concentration; guilty feelings; suicidal ideas. |
| Eating Disorder | intense fear of gaining weight; binge and purge; rumination; weight and appetite change. |
| OCD | obsession; compulsion. |
| PTSD | often develops after a shocking, dangerous event; flashbacks; bad dreams. |

Table 6: 7 mental disorders detected in our work with their typical symptoms.

## B Detailed Data Statistics

For the multiple MDD datasets (Section 3.1), we show the number of users suffering from each disease in Table 7. The distribution of the 7 diseases are similar to SMHD (Cohan et al., 2018), and the training/validation/testing set is 8:1:1.

In addition, the users in these two datasets can suffer from multiple mental disorders simultaneously. So we also provide the distribution of the number of diseases on a single user in Table 8. The statistic shows that 57% users suffer from more

| Disease | # Users (Reddit) | # Users (Twitter) |
|---------|------------------|-------------------|
| Depression | 3105 | 1448 |
| Anxiety | 2239 | 933 |
| ADHD | 2374 | 1370 |
| Bipolar Disorder | 1366 | 2011 |
| OCD | 753 | 368 |
| PTSD | 391 | 588 |
| Eating Disorder | 138 | 0 |

Table 7: Number of users suffering from each disease in the multiple MDD datasets (Reddit and Twitter).

than one mental disorders in the Reddit dataset, and PsyEx can achieve superior performance for its specific design targeting these comorbidity scenarios.

| # Disease | # Users (Reddit) | # Users (Twitter) |
|-----------|------------------|-------------------|
| 1 | 2326 | 2842 |
| 2 | 1738 | 1716 |
| 3 | 931 | 134 |
| 4 | 407 | 8 |
| 5 | 152 | 1 |
| 6 | 51 | 0 |
| 7 | 14 | 0 |

Table 8: Distribution of a user's disease counts. For example, there are 1738 users suffering from two mental diseases in the Reddit dataset.

## C Detailed Experimental Settings of Baselines

For the CNN backbone of *BERT* and *Symp* method, the model structure is the same as that of Nguyen et al. (2022). We train both model with batch size=64, but we set the learning rate as 0.01 when using symptom features, and as 0.003 when using BERT embedding. The *HAN-GRU* model is trained with batch size=32 and learning rate=0.0001. The posting list will be truncated to preserve the earliest 256 posts at most. The average posting number of users in the dataset is 115.2, so 256 is safe enough to preserve almost all the posts of a user.

In the ablation test of different risky post screening methods, the sentence BERT model we utilized for *Similarity* and *K-Means* is paraphrase-MiniLM-L6-v2. What's more, the *Similarity* (Zhang et al., 2022a) method extracts key posts according to the cosine similarity between post and mental disease descriptions, which are the description of 38 symptoms (see Table 12) manually summarized from DSM-5.

Further, to infer the symptoms mentioned in the posts, we perform some pre-processing steps. First,

| Model | Depression | Anxiety | ADHD | Bipolar | OCD | PTSD | Eating | Avg. F1 |
|---|---|---|---|---|---|---|---|---|
| Two-Stream PsyEx | 87.89 | **89.84** | 84.4 | **91.58** | **81.69** | **81.69** | 85.71 | **86.12** |
| w/o symp-stream | **88.12** | 89.09 | **84.55** | 89.33 | 76.71 | 81.58 | 83.33 | 84.67 |
| w/o multi-attn | 87.64 | 88.64 | 82.84 | 89.7 | 76.39 | 79.49 | 81.08 | 83.68 |
| w/o multi-task | 87.69 | 88.89 | 84.21 | 91.16 | 75 | 84.21 | **86.67** | 85.40 |

Table 9: Ablation tests for the design choices of PsyEx, reporting F1 score of each disease (detailed results of Table 3).

| Screen method | Depression | Anxiety | ADHD | Bipolar | OCD | PTSD | Eating | Avg. F1 |
|---|---|---|---|---|---|---|---|---|
| Symptom-based (PsyEx) | **87.89** | **89.84** | **84.4** | **91.58** | **81.69** | **81.69** | **85.71** | **86.12** |
| Similarity | 86.84 | 87.7 | 83.03 | 87.84 | 78.26 | 77.14 | 76.47 | 82.47 |
| K-Means | 79.34 | 81.15 | 68.66 | 82.0 | 70.34 | 73.42 | 68.42 | 74.76 |
| Last | 69.27 | 67.89 | 55.31 | 67.6 | 45.8 | 46.15 | 45.16 | 56.74 |

Table 10: Ablation tests for different risky post screening methods, reporting F1 score of each disease (detailed results of Table 4)

we use *blingfire*[17] to split a post into sentences. Then, we use regular expressions to filter out the hyperlink format like "[anchor text] (web url)" and preserve the anchor text. Finally, we remove sentences like "[removed]".

## C.1 ChatGPT baseline

Due to the input length limitation of ChatGPT, we utilize the screening method described in Sec. 2.1, which selects 16 posts from the user history instead of using all of them.

To maximize the effectiveness of ChatGPT, we made substantial efforts in prompt engineering. Through this process, we observed that directly requesting prediction results yielded unsatisfactory performance in binary setting. To address this, we introduced the requirement for ChatGPT to provide explanations alongside its predictions, leading to a significant improvement in accuracy, especially eating disorder (See Table 11). We also attempted to include explanation instructions in a multi-label setting; however, the outcome was even worse than without explanations. This suggests that there is still a long way to go in effectively utilizing Chat-GPT for this task, and we hope to address this in the future work.

We present the final prompts for both binary and multi-label settings below.

**Binary setting**

> Please predict whether the user has mental disorders, such as DISEASE, from the clues showed in his/her posts on reddit. Your answer should be a single 'yes' or 'no', followed by an explanation of the predicted result.

[17]https://github.com/microsoft/BlingFire

**Multi-label setting**

> Please predict the user's mental disorders according to his/her posts on reddit. The mental disorders should be chosen from: 'depression', 'anxiety', 'bipolar', 'eating disorder', 'ADHD', 'PTSD', 'OCD'. The user can have multiple disorders, or have no disorder. Your answer format should be: 'mental disorders: {MENTAL DISORDERS}'.

## D Detailed Ablation Results

We show the detailed results of ablation tests in Table 9 and Table 10. Without symptom stream (i.e., w/o symp-stream) or disease-specific attention layer (i.e., w/o multi-attn), the F1 score on nearly all the diseases dropped, especially the rarer diseases like eating disorder and OCD. What's more, we can notice a significant drop in the performance without symptom-based risky post screening in all the diseases, suggesting the importance of a precise screening method to filter out the noisy data.

### D.1 Impact of the Number of Selected Posts

Here we study the impact of the number of posts selected in risky posts screening (see Figure 5). We can observe that 16 posts have the best performance for nearly all the diseases. To find out the reason, we calculate the average symptom probability of the selected $K$ posts sorted by its highest symptom probability, which is 0.25 for the posts ranked 16, meaning that 16 posts is enough to include most of the symptomatic posts and adding more posts can easily introduce some noisy data.

## E Psychiatric Symptoms

We use serial numbers to represent symptoms in Figure 3 and Figure 4, so we provide the corre-

| Method | Depression | Anxiety | ADHD | Bipolar | OCD | PTSD | Eating | Avg. F1 |
|---|---|---|---|---|---|---|---|---|
| direct | 66.92 | 70.16 | 59.36 | 67.95 | 39.22 | 53.13 | 8.7 | 52.21 |
| explanation | 70.12 | 73.09 | 64.08 | 67.12 | 52.98 | 67.61 | 29.73 | 60.68 |

Table 11: Mental disease detection results across 7 diseases utilizing ChatGPT in binary setting

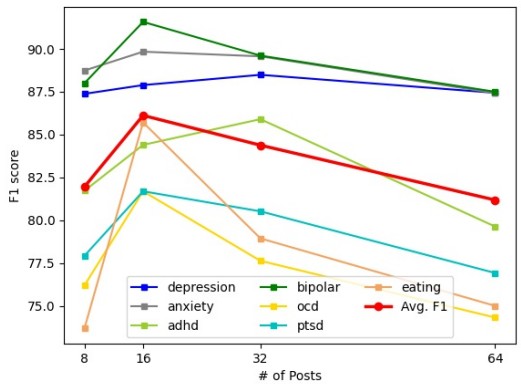

Figure 5: Impact of the number of selected posts on each disease and the mean F1.

sponding symptom names in Table 12 for reference.

These symptoms are defined in the symptom identification dataset proposed by Zhang et al. (2022b). This dataset contains 8,554 sentences extracted from the Reddit corpus. All the sentences are annotated with symptoms that can be identified from the sentence. The average Fleiss's $\kappa$ for all the 38 symptoms is 0.7708.

| id | Symptom |
|---|---|
| 1 | Anxious Mood |
| 2 | Autonomic symptoms |
| 3 | Cardiovascular symptoms |
| 4 | Catatonic behavior |
| 5 | Decreased energy tiredness fatigue |
| 6 | Depressed Mood |
| 7 | Gastrointestinal symptoms |
| 8 | Genitourinary symptoms |
| 9 | Hyperactivity agitation |
| 10 | Impulsivity |
| 11 | Inattention |
| 12 | Indecisiveness |
| 13 | Respiratory symptoms |
| 14 | Suicidal ideas |
| 15 | Worthlessness and guilty |
| 16 | Avoidance of stimuli |
| 17 | Compensatory behaviors to prevent weight gain |
| 18 | Compulsions |
| 19 | Diminished emotional expression |
| 20 | Do things easily get painful consequences |
| 21 | Drastic shift in mood and energy |
| 22 | Fear about social situations |
| 23 | Fear of gaining weight |
| 24 | Fears of being negatively evaluated |
| 25 | Flight of ideas |
| 26 | Intrusion symptoms |
| 27 | Loss of interest or motivation |
| 28 | More talkative |
| 29 | Obsession |
| 30 | Panic fear |
| 31 | Pessimism |
| 32 | Poor memory |
| 33 | Sleep disturbance |
| 34 | Somatic muscle |
| 35 | Somatic symptoms others |
| 36 | Somatic symptoms sensory |
| 37 | Weight and appetite change |
| 38 | Anger Irritability |

Table 12: Id and its corresponding symptoms