# OpenReview forum: "Detection of Multiple Mental Disorders from Social Media with Two-Stream Psychiatric Experts"
_EMNLP/2023/Conference — EMNLP 2023 Main_

### Official Review · Reviewer_CNuF · 2023-07-23

**Typos Grammar Style And Presentation Improvements:** 1.add a paragraph to describe the exp…
**Soundness:** 3

**Excitement:**

3: Ambivalent: It has merits (e.g., it reports state-of-the-art results, the idea is nice), but there are key weaknesses (e.g., it describes incremental work), and it can significantly benefit from another round of revision. However, I won't object to accepting it if my co-reviewers champion it.

**Paper Topic And Main Contributions:**

The article proposes a novel framework called PsyEx, which combines symptom-based screening with psychiatric knowledge explicitly and implicitly embedded into the risky post stream. The proposed framework is evaluated on a large-scale dataset of social media posts, and the results show that it outperforms several baseline models in detecting multiple mental disorders.

**Questions For The Authors:**

The authors need to verify this on more data with more state-of-the-art methods.
1.Limited scope of population and Lack of clinical real cohort: The proposed framework is evaluated on a single dataset, which may not be representative of the general population.

2. Lack of explainability. The paper does not provide a detailed discussion of the interpretability of the model. Although the model is excellent at detecting mental disorders, its inner workings are still not well understood. Therefore, the paper may need to discuss in more detail how to improve the explainability of the model and explain how the model makes predictions.

3.Lack of parameter details: The article provides a detailed theoretical description of the proposed approach, but does not provide much information on parameters in practice.

**Reasons To Accept:**

1. Novel approach: This article presents a novel approach to mental disease detection that considers the possibility of multiple disorders occurring simultaneously and incorporates domain knowledge for more interpretable results.

2. Improved detection performance: The two-stream architecture of this approach improves detection performance by complementing symptom-based screening with psychiatric knowledge explicitly and implicitly embedded into the risky post stream.

3. Applicability to rare classes: This approach brings significant improvement to rarer diseases on which the baselines struggled heavily, addressing the issue of rare classes in mental disease detection.

**Reasons To Reject:**

1.Limited scope of population and Lack of clinical real cohort: The proposed framework is evaluated on a single dataset, which may not be representative of the general population.

2. Lack of explainability. The paper does not provide a detailed discussion of the interpretability of the model. Although the model is excellent at detecting mental disorders, its inner workings are still not well understood. Therefore, the paper may need to discuss in more detail how to improve the explainability of the model and explain how the model makes predictions.

3.Lack of parameter details: The article provides a detailed theoretical description of the proposed approach, but does not provide much information on parameters in practice.

**Reproducibility:**

4: Could mostly reproduce the results, but there may be some variation because of sample variance or minor variations in their interpretation of the protocol or method.

**Reviewer Confidence:**

4: Quite sure. I tried to check the important points carefully. It's unlikely, though conceivable, that I missed something that should affect my ratings.

---

> ### Author Rebuttal · Authors · 2023-08-28
>
> 1. The authors need to verify this on more data with more state-of-the-art methods. Limited scope of population and Lack of clinical real cohort: The proposed framework is evaluated on a single dataset, which may not be representative of the general population.
>
> **Ans:** In our paper, we utilized the SMHD dataset, which includes real social media posts from 5624 diagnosed users and 17209 control users. This dataset covers 7 common mental disorders, making it one of the most comprehensive datasets in terms of mental disorder detection task.
>
> Further, to address your concern, we observed the potential to extend our method to different types of social media platforms, such as Twitter. Therefore, we conducted supplementary experiments on a Twitter dataset[1], which encompasses eight distinct mental disorders. Importantly, 60% of it is hand-annotated, resulting in precise disorder labels. We worked with a subset of this Twitter dataset, and the corresponding data statistics are outlined below:
>
>
> |            | depression | anxiety | adhd | bipolar | ocd | ptsd | control |
> |------------|------------|---------|------|---------|-----|------|---------|
> | # of users | 1448       | 933     | 1370 | 2011    | 368 | 588  | 4804    |
>
> |            | 1 disorder | 2 disorders | 3 disorders | 4 disorders | 5 disorders |
> |------------|------------|-------------|-------------|-------------|-------------|
> | # of users | 2842       | 1716        | 134         | 8           | 1           |
>
> In this twitter dataset, each user averages 8821.1 tweets, with each tweet containing an average of 13.59 words. This indicates a higher frequency of shorter content compared to Reddit posts.
>
> We also introduce an additional baseline, referred to as "Depression-RoBERTa"[2], to demonstrate the performance of fine-tuning a pre-trained RoBERTa model in this multiple disorder detection task.
>
> We directly apply the hyper-parameter settings (e.g., batch size, learning rate, etc.) of each model from the SMHD dataset without further tuning (Refer to Section 3.3 and Appendix C for details). As each user's tweets are shorter yet more frequent than Reddit posts, we select K=64 tweets as risky posts for PsyEx (K=16 in the SMHD dataset). The experimental outcomes on the Twitter dataset are as follows:
>
> |                    | depression | anxiety   | adhd      | bipolar   | ocd       | ptsd      | Mean F1   |
> |--------------------|------------|-----------|-----------|-----------|-----------|-----------|-----------|
> | tf-idf+LR          | 36.5       | 37.33     | 61.19     | 56.01     | 3.8       | 8.5       | 33.89     |
> | Symp               | 41.49      | 37.93     | 31.24     | 41.52     | 32.49     | 22.46     | 34.52     |
> | BERT               | 56.3       | 53.43     | 57.86     | 62.38     | 44.24     | 45.57     | 53.30     |
> | Depression-RoBERTa | 58.29      | 53.83     | 61.32     | 62.78     | 46.69     | 49.38     | 55.38     |
> | HAN-GRU            | 72.24      | 71.21     | 65.17     | 75.44     | 58.58     | 60.24     | 67.15     |
> | PsyEx              | **76.51**  | **77.31** | **67.15** | **78.36** | **67.35** | **65.64** | **72.05** |
>
> The results demonstrate the effectiveness of our proposed PsyEx model on Twitter data, surpassing the performance of all baselines. This illustrates the model's capability to generalize across various data types.
>
>
> [1] Suhavi et al. “Twitter-STMHD: An Extensive User-Level Database of Multiple Mental Health Disorders.” International Conference on Web and Social Media (2022).
>
> [2] Poswiata et al. “OPI@LT-EDI-ACL2022: Detecting Signs of Depression from Social Media Text using RoBERTa Pre-trained Language Models.” LTEDI (2022).
>
>
> 2. Lack of explainability. The paper does not provide a detailed discussion of the interpretability of the model. Although the model is excellent at detecting mental disorders, its inner workings are still not well understood. Therefore, the paper may need to discuss in more detail how to improve the explainability of the model and explain how the model makes predictions.
>
> **Ans:** The explainability in our method can be mainly derived from two components: the symptom features and the multiple attention layers, which function as dedicated "expert" for each disorder.
>
> First, in Symptom-based Risky Posts Screening (Sec 2.1), we predicted the probabilities of any symptoms reflected in one post, so that we can detect the symptoms that the user may be suffering, and find highly risky posts that shows signals of certain symptom. The symptoms then become direct features for model prediction.
>
> Second, the attention mechanism used in our model (Eq. 2,3) allows us the check the importance(alpha) of each post/symptom in deciding the final prediction, so that we know which posts/symptoms have major effects on the model prediction.
>
> In Section 3.5, we further combine the 2 aspects and calculate symptom contribution vector. This vector further helps us visualize the contribution of each symptom to the prediction of each disease according to the model (Figure 3).
>
> 3. Lack of parameter details: The article provides a detailed theoretical description of the proposed approach, but does not provide much information on parameters in practice.
>
> **Ans:** Thank you for pointing this out. We have provided descriptions of the experimental parameter settings in Section 3.2 and Appendix C, such as learning rate and batch size. Here we further supplement the specification of the model's parameter. We don't know ChatGPT's exact number of parameter, so we approximate it with its ancestor GPT-3.
>
> | Model                          | # of Parameter |
> |--------------------------------|----------------|
> | Symp                           | 38.5K          |
> | BERT                           | 110M           |
> | **PsyEx(bert-tiny)**           | **4.8M**       |
> | **PsyEx(mental-bert-uncased)** | **123M**       |
> | GPT-3                          | 175B           |
>
> As shown in the table above, the parameter of PsyEx(mental-bert-uncased) is similar to the commonly used BERT model, and PsyEx(bert-tiny) is much more smaller. They are also significantly smaller than recent LLMs like GPT-3.

---

### Official Review · Reviewer_qJms · 2023-08-07

**Soundness:** 3

**Excitement:**

4: Strong: This paper deepens the understanding of some phenomenon or lowers the barriers to an existing research direction.

**Paper Topic And Main Contributions:**

Authors propose a model to detect multiple mental disorders from social media (reddit) posts. They use two streams of learning that should understand if the post is involving a symptom related to one of the diseases and if the post is risky (ie. relevant to the mental health) or not. Mental health comorbidities are common and there is not much work on detecting multiple diseases that person can have in the same time, while that is pretty common in practice. Authors use hierarchical network to learn post embedding in the lower level and then user embedding given all relevant user posts. According to evaluation, the proposed model performs better than strong existing baselines, including ChatGPT.

**Questions For The Authors:**

Is the dataset going to be public? Making it public (together with symptoms and disease labels) would bring a lot to mental health research.

**Reasons To Accept:**

1. The application of the paper is relevant and very important.
2. Authors have developed method to automatically extract symptoms, labels and understand persons diseases from reddit posts
3. Proposed model performance is better than strong baselines and something that is generally hard to achieve on mental health data given the skewed datasets. This is especially true for multi-label settings.

**Reasons To Reject:**

1. It is not clear if this can be applied to shorter comments or different social media (twitter, facebook)
2. To use or extend the model, psychiatric knowledge is required within the team in order to be able to understand all symptoms and how are they related to dieseases.

**Reproducibility:**

3: Could reproduce the results with some difficulty. The settings of parameters are underspecified or subjectively determined; the training/evaluation data are not widely available.

**Reviewer Confidence:**

5: Positive that my evaluation is correct. I read the paper very carefully and I am very familiar with related work.

---

> ### Author Rebuttal · Authors · 2023-08-28
>
> 1. It is not clear if this can be applied to shorter comments or different social media (twitter, facebook)
>
> **Ans:** Thank you for the insightful question. Obtaining suitable datasets for the detection of multiple mental disorders is quite challenging, as most datasets tend to focus on the detection of single disorders (e.g., depression), or doesn't have open access. Fortunately, we came across a recent Twitter dataset[1] encompassing users with eight distinct mental disorders. A notable aspect of this dataset is that 60% of it is hand-annotated, resulting in precise disorder labels. We worked with a subset of this Twitter dataset, and the data statistics are as follows:
>
> |            | depression | anxiety | adhd | bipolar | ocd | ptsd | control |
> |------------|------------|---------|------|---------|-----|------|---------|
> | # of users | 1448       | 933     | 1370 | 2011    | 368 | 588  | 4804    |
>
> |            | 1 disorder | 2 disorders | 3 disorders | 4 disorders | 5 disorders |
> |------------|------------|-------------|-------------|-------------|-------------|
> | # of users | 2842       | 1716        | 134         | 8           | 1           |
>
> In this dataset, each user averages 8821.1 tweets, with each tweet containing an average of 13.59 words. This indicates a higher frequency of shorter content compared to Reddit posts.
>
> We conduct experiments on this twitter dataset in binary setting. We directly apply the hyper-parameter settings (e.g., batch size, learning rate, etc.) of each model from the SMHD dataset without further tuning (Refer to Section 3.3 and Appendix C for details). As each user's tweets are shorter yet more frequent than Reddit posts, we select K=64 tweets as risky posts for PsyEx (K=16 in the SMHD dataset). The experimental outcomes on the Twitter dataset are as follows:
>
> |                    | depression | anxiety   | adhd      | bipolar   | ocd       | ptsd      | Mean F1   |
> |--------------------|------------|-----------|-----------|-----------|-----------|-----------|-----------|
> | tf-idf+LR          | 36.5       | 37.33     | 61.19     | 56.01     | 3.8       | 8.5       | 33.89     |
> | Symp               | 41.49      | 37.93     | 31.24     | 41.52     | 32.49     | 22.46     | 34.52     |
> | BERT               | 56.3       | 53.43     | 57.86     | 62.38     | 44.24     | 45.57     | 53.30     |
> | HAN-GRU            | 72.24      | 71.21     | 65.17     | 75.44     | 58.58     | 60.24     | 67.15     |
> | PsyEx              | **76.51**  | **77.31** | **67.15** | **78.36** | **67.35** | **65.64** | **72.05** |
>
> The results demonstrate the effectiveness of our proposed PsyEx model on Twitter data, surpassing the performance of all baselines. This illustrates the model's capability to generalize across various data types.
>
> [1] Suhavi et al. “Twitter-STMHD: An Extensive User-Level Database of Multiple Mental Health Disorders.” International Conference on Web and Social Media (2022).
>
>
> 2. To use or extend the model, psychiatric knowledge is required within the team in order to be able to understand all symptoms and how they are related to dieseases.
>
> 	**Ans:** We agree that psychiatric knowledge is vital to our approach, endowing PsyEx with high precision and interpretability based on a relatively higher bar of usage. However, we expect that organizations who would deploy such disease detection system for social good should at least have some members with psychiatric knowledge, so that they can audit the model predictions and provide further psychiatric aid. Consequently, the requirement of psychiatric knowledge would not be a real obstacle.
>
> 3. Is the dataset going to be public? Making it public (together with symptoms and disease labels) would bring a lot to mental health research.
>
> 	**Ans:** Yes, we will make the dataset and code be public.

---

### Official Review · Reviewer_hhgj · 2023-08-12

**Soundness:** 4

**Excitement:**

4: Strong: This paper deepens the understanding of some phenomenon or lowers the barriers to an existing research direction.

**Paper Topic And Main Contributions:**

The authors presented a new approach towards the Mental Disorder Detection task. They presented a thorough experimental analysis and showed the superiority of their own approach.

**Questions For The Authors:**

How the task fine-tuned models perform in this scenario? Do they outperform your approach?

**Reasons To Accept:**

The approach using two stream psychiatric experts is novel. A detailed experimental analysis was presented. The results seem to improve upon the SOTA models.

**Reasons To Reject:**

Multiple SOTA task fine-tuned models were not included in the experiments (i.e. Mental-BERT, Depression-RoBERTa).

**Reproducibility:**

3: Could reproduce the results with some difficulty. The settings of parameters are underspecified or subjectively determined; the training/evaluation data are not widely available.

**Reviewer Confidence:**

4: Quite sure. I tried to check the important points carefully. It's unlikely, though conceivable, that I missed something that should affect my ratings.

---

> ### Author Rebuttal · Authors · 2023-08-28
>
> **Q1.** How the task fine-tuned models (Mental-BERT, Depression-RoBERTa) perform in this scenario? Do they outperform your approach?
>
> **Ans:** Thank you for raising this point. In our PsyEx model, we have already incorporated Mental-BERT as the post encoder. Directly finetuning Mental-BERT[1] or Depression-RoBERTa[2] is virtually the ablation of the proposed method.
>
> The outcomes from direct task fine-tuned models fall behind PsyEx, as indicated in the table below. This can be attributed to these models failed to capture the sequential structure of the posts. Therefore, design choices like the Two-Stream Psychiatric Experts Model (Sec. 2.2) in the proposed PsyEx model are crucial to the model's significantly improved performance.
>
>
> |                           | depression | anxiety | adhd  | bipolar | ocd   | ptsd  | eating | Mean F1 |
> |---------------------------|------------|---------|-------|---------|-------|-------|--------|---------|
> | MentalBERT(base)          | 65.05      | 61.68   | 54.54 | 62.54   | 45.07 | 47.5  | 37.84  | 53.46   |
> | Depression-RoBERTa(large) | 63.46      | 62.3    | 53.55 | 63.49   | 48.73 | 52.78 | 53.85  | 56.88   |
> | PsyEx                     | 87.89      | 89.84   | 84.4  | 91.58   | 81.69 | 81.69 | 85.71  | 86.12   |
>
>
> [1] Ji et al. "MentalBERT: Publicly Available Pretrained Language Models for Mental Healthcare" LREC (2022).
>
> [2]Poswiata et al. “OPI@LT-EDI-ACL2022: Detecting Signs of Depression from Social Media Text using RoBERTa Pre-trained Language Models.” LTEDI (2022).

---

### Meta-Review · Area_Chair_Cb6i · 2023-09-18

**Recommendation:** 5

**Metareview:**

This paper tackles the problem of detecting multiple mental health disorders from social media posts, which is experimented with Reddit posts. The proposed method leverages two streams of learning to enable this detection.

The contribution is novel, achieving remarkable improvements over competitive baseline models on a challenging and important task.

I appreciate the authors for the effort in preparing a solid rebuttal and I would likewise ask them to incorporate the new results reported in the rebuttals in a revised version of the paper.

---

### Decision · Program_Chairs · 2023-10-07

**Decision:**

Accept-Main

**Comment:**

This paper tackles the problem of detecting multiple mental health disorders from social media posts, which is experimented with Reddit posts. The proposed method leverages two streams of learning to enable this detection.

The contribution is novel, achieving remarkable improvements over competitive baseline models on a challenging and important task.

I appreciate the authors for the effort in preparing a solid rebuttal and I would likewise ask them to incorporate the new results reported in the rebuttals in a revised version of the paper.